DATA RELEASE

# MEMO: Monitoring of exotic mosquitoes in Belgium

Isra Deblauwe[1], Dimitri Brosens[2,*], Katrien De Wolf[1,3], Nathalie Smitz[4], Adwine Vanslembrouck[1], Anna Schneider[1], Jacobus De Witte[1], Ingrid Verlé[1], Wouter Dekoninck[5], Marc De Meyer[4], Thierry Backeljau[5,6], Sophie Gombeer[5], Kenny Meganck[4], Ann Vanderheyden[4], Ruth Müller[1] and Wim Van Bortel[1,7]

1 Unit Entomology, Dept. of Biomedical Sciences, Institute of Tropical Medicine (ITM), Nationalestraat 155, 2000, Antwerpen, Belgium
2 Research Institute for Nature and Forest (INBO), Havenlaan 88 b73, 1000, Brussels, Belgium
3 Terrestrial Ecology Unit, Dept. of Biology, Ghent University, Ghent, Belgium
4 Royal Museum for Central Africa (RMCA - BopCo), Leuvensesteenweg 17, 3080 Tervuren, Belgium
5 Royal Belgian Belgian Institute for Natural Sciences (RBINS - BopCo & Scientific Heritage Service), Vautierstraat 29, 1000, Brussels, Belgium
6 Evolutionary Ecology Group, University of Antwerp, Universiteitsplein 1, 2610 Antwerp, Belgium
7 Outbreak Research team, Institute of Tropical Medicine (ITM), Nationalestraat 155, 2000 Antwerp, Belgium

## ABSTRACT

'MEMO: Monitoring of Exotic MOsquitoes in Belgium' is a sampling event dataset published by the Institute of Tropical Medicine (ITM) in Antwerp, Belgium. It forms part of the early detection of exotic mosquito species (EMS) along high-risk introduction routes in Belgium, where data are collected at defined points of entry (PoEs) using a standardised protocol. The MEMO dataset contains mosquito sampling counts performed between 2017 and 2020. MEMO+2020, an extension of the MEMO dataset, contains only *Aedes albopictus* mosquito trap counts performed in 2020. Here, we present these data published as a standardised Darwin Core archive, which includes, for each sampling event, an eventID, date, location and sampling protocol (in the event core); and an occurrenceID for each occurrence (tube), the number of collected individuals per tube, species status (present/absent), information on the identification and scientific name (in the occurrence extension).

**Subjects** Ecology, Biodiversity, Taxonomy

**Submitted:** 30 March 2022

* Corresponding author. E-mail: dimitri.brosens@inbo.be

Preprint submitted at https://doi.org/10.5281/zenodo.6334125

Included in the series: *Vectors of human disease* (https://doi.org/10.46471/GIGABYTE_SERIES_0002)

*Gigabyte*, 2022, 1–**??**

## DATA DESCRIPTION

Following previous exotic mosquito species (EMS) surveillance projects in Belgium [1], a 3-year national active EMS monitoring project 'MEMO: Monitoring of Exotic MOsquitoes in Belgium' started in July 2017. MEMO forms part of the early detection of exotic mosquito species (EMS) along high-risk introduction routes in Belgium, where data are collected at defined points of entry (PoEs) using a standardised protocol.

Here, we present the MEMO sampling event dataset [2, 3], published by the Institute of Tropical Medicine (ITM) in Antwerp, Belgium, which contains mosquito sampling counts performed between 2017 and 2020. In addition, the MEMO+2020 dataset [4, 5] is an extension of the MEMO dataset that contains only *Aedes albopictus* mosquito trap counts performed in 2020.

**Figure 1.** Overview of the MEMO points of entry (PoEs). Blue dots: MEMO; red diamonds: MEMO+2020. In total, 33 locations were sampled in MEMO and six locations in MEMO+2020.

## Context

The early detection of exotic mosquito species (EMS), such as *Aedes albopictus* (NCBI:txid7160), *Aedes japonicus* (NCBI:txid140438), *Aedes koreicus* (NCBI:txid586676) and *Aedes aegypti* (NCBI:txid7159), is of paramount importance, especially along high-risk introduction routes (points of entry; PoEs) before populations become established to prevent local transmission of mosquito-borne diseases.

## METHODS

### MEMO (dataset 1) [2, 3]

In 2017, 2018 and 2019, active monitoring was implemented by researchers from ITM in 20–23 different PoEs (Figure 1) with exact GPS coordinates. The risk of introduction and establishment of EMS at each PoE was re-evaluated annually to ensure that monitoring focused on the highest risk sites. Different sampling protocols were used, defined, planned and coordinated by ITM researchers, including BG-Sentinel, Mosquito Magnet® and gravid traps to collect host-seeking female mosquitoes; oviposition traps to detect eggs; and larval sampling [6]. At each PoE, a combination of different trapping methods was used simultaneously.

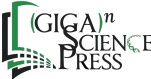

The collected specimens were sorted and identified using morphological characteristics [7–10]. The caught EMS, as well as 5% of all collected mosquitoes, were identified using DNA-based techniques to validate and confirm the morphological identification. Tissue and DNA were subsequently deposited in a molecular reference collection hosted at the Royal Belgian Institute of Natural Sciences (RBINS) [2].

A specific DNA-based identification pipeline was developed [11] to enable accurate identification to species (or biotype for *Culex pipiens* s.s.) level of all mosquitoes occurring in Belgium (native and potential EMS). Further, a morphological collection with a representation of 23 species and the most intact specimens sampled during the MEMO project was generated for future reference and is also hosted at RBINS [3]. Data management was done using VECMAP® software [12] (Avia-GIS, Zoersel, Belgium).

### MEMO+2020 (dataset 2) [4, 5]

In 2020, active monitoring was implemented at six different PoEs (Figure 1). The focus was on parking lots along the highway as this pathway for exotic *Aedes* species is becoming more important. Four fixed parking lots (Aische-en-Refail, Raeren, Marke, Saint-Ghislain) were monitored at first, but following the detection of *Ae. albopictus* in the Netherlands, two further parking lots (Minderhout, Gierle) were added. Oviposition traps were used to collect eggs, and potential breeding sites were inspected for larvae [13]. MEMO+2020 sampling resulted in one positive identification of *Ae. albopictus*.

### Taxonomic coverage of the two datasets

In the period 2000–2009, substantial changes were proposed for the Aedini tribe taxonomy, which resulted in almost tripling the number of genera in the entire Culicidae family. A recent publication [14] proposed a return to the taxonomy from before 2000, restoring a classification system useful for the operational community. This latter classification system was used during the MEMO and MEMO+2020 projects Most specimens collected of the *Anopheles maculipennis* complex were molecularly identified up to species level. Three adults could morphologically be identified as *Culex torrentium*, other specimens are grouped together with *Culex pipiens* s.l. (Figure 2).

### Geographic coverage of the two datasets

Belgium is a small country in Western Europe. To the west, its 70-km coastline fronts the North Sea; to the north lies the Netherlands; to the east, Germany, and to the south, France and Luxembourg. Biogeographically, the fauna of eastern Belgium belongs to the Central European province of the Eurasian (Palaearctic), Continental Biogeographical Region. By contrast, the rest of the country primarily comprises Atlantic fauna (Atlantic Biogeographical Region) (Figure 3).

Politically and geographically, the country is divided into three parts: Flanders, Wallonia and the Brussels Capital Region. In Flanders (13,522 km$^2$ with a population of about 6 million people), to the north, soils are mainly sandy to loamy. The Brussels Capital Region is a small region (162 km$^2$) entirely situated in the sandy loam area. In Wallonia (17,006 km$^2$ and about 3.5 million people), to the south, soils and habitats are more diverse, ranging from forests to rocky and calcareous grasslands on loam and chalky soils. Eastern Wallonia, near the German border, includes the Hautes Fagnes, a large area of bogs and peat.

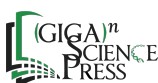

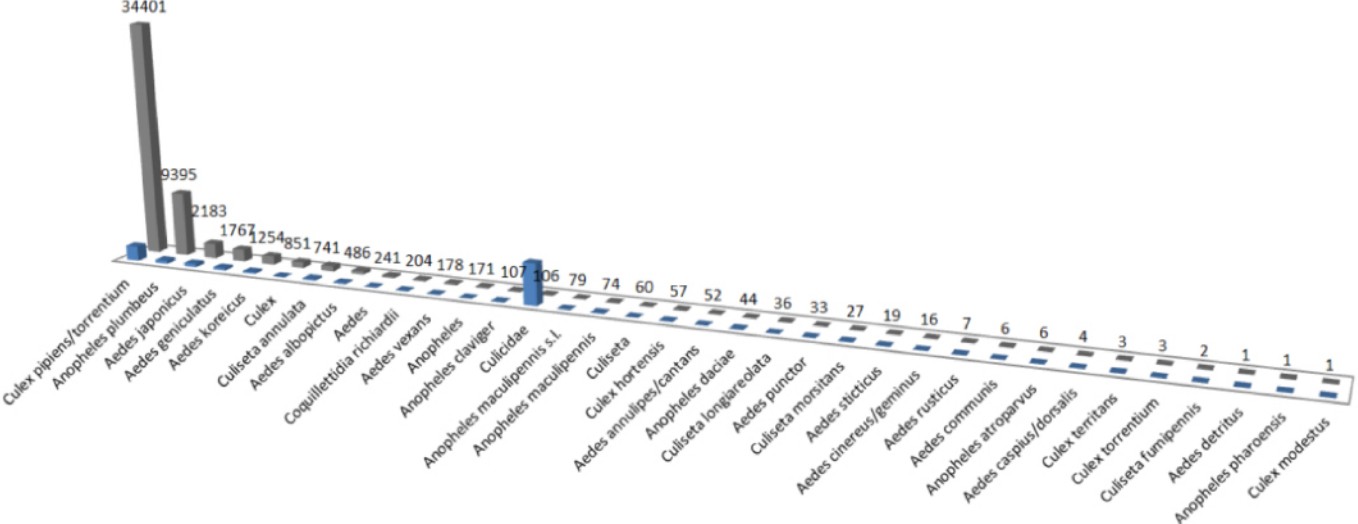

**Figure 2.** Morphological identifications and the number of individuals (gray) and records (blue) in the MEMO dataset.

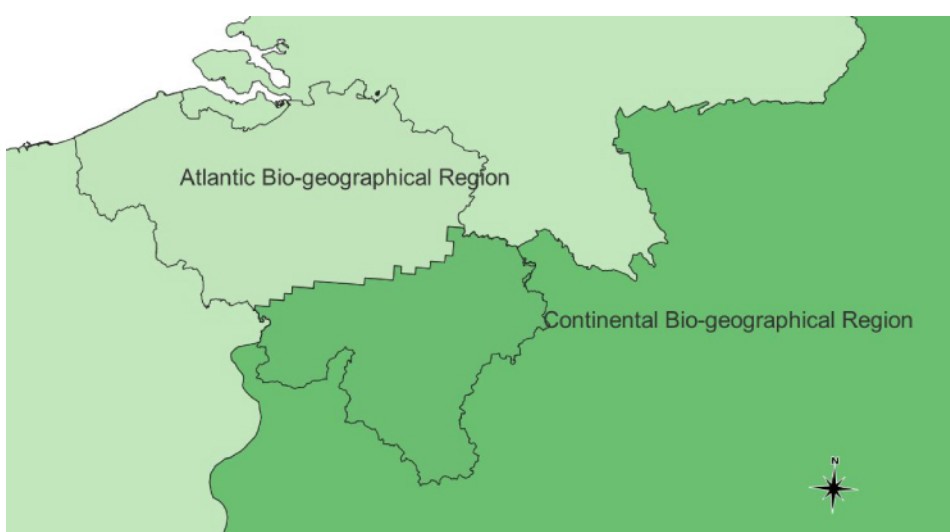

**Figure 3.** Biogeographical areas of Belgium.

Belgian has a temperate maritime climate that is influenced by the North Sea and the Atlantic Ocean with substantial precipitation in all seasons. Summers are moderate and winters are mild.

## Temporal coverage

The first phase of the project ran from July 2017 until June 2020 (MEMO dataset), and the second phase from July 2020 until November 2020 (MEMO+2020 dataset).

The overall distribution of the mosquito sampling events over time is illustrated in Figure 4.

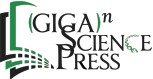

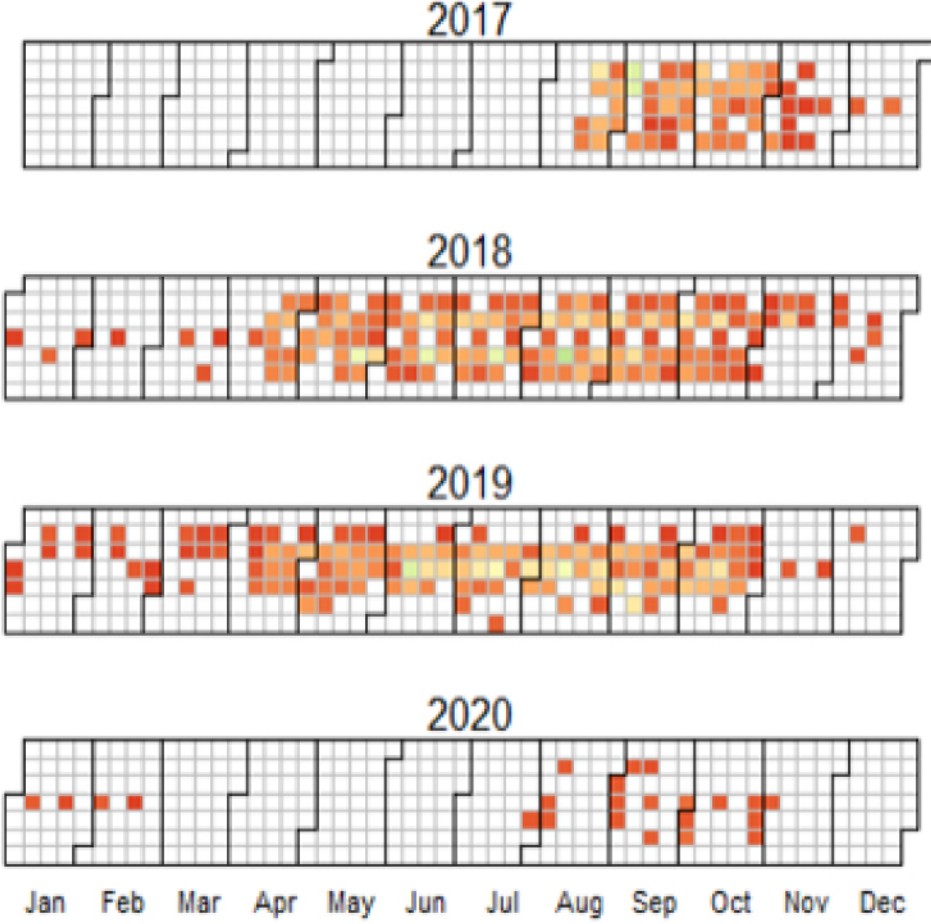

**Figure 4.** Temporal coverage of the sampling effort. Calendar heatmap of number of records. Red is intensive sampling.

## Sample processing for the two datasets

On arrival at the laboratory, adult specimens were killed by freezing at −20 °C. Larvae were transported alive to the laboratory and killed by a thermal shock with hot water (70 °C). Next, larvae were transferred in 80% ethanol for preservation before morphological identification. In the case of EMS, larvae were then transferred in absolute ethanol for further DNA-based species validation. Morphological identification of adults and larvae was done with a stereomicroscope using dichotomic and digital keys [7–10]. DNA-barcoding technology [15] was applied to validate the morphological identification of EMS, of 5% of the annual sampling (quality control), and to identify damaged adults and larvae or species complexes (following the methodology of Smitz *et al.* [16]). The step-by-step procedures can be viewed in the Github MEMO repository [11].

Polystyrene pieces collected from oviposition traps were checked for EMS eggs in the laboratory using a stereomicroscope. A subsample of the eggs from the polystyrene piece (1–5 eggs per side) was always DNA barcoded [17]. In 2020, the positive polystyrene pieces were immersed in water in secured containers, which were stored in a climate-controlled cupboard. Hatched larvae (third or fourth instar) were stored in absolute ethanol for

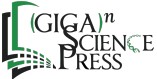

morphological identification. In case the eggs did not hatch, eggs were DNA barcoded for species identification.

The generated DNA barcodes were deposited in the open access GenBank sequence database, maintained by the National Center for Biotechnology Information (NCBI), linked to BioProject ID PRJNA837425 [16–20]. The generated microsatellites database for *Aedes japonicus* population genetic investigation [17] was deposited on the open-access Dryad Digital Repository [21].

Pinned adults and mounted larvae specimens are stored in the collections of the RBINS (Collection Identifier: IG32776; RBINS: IG34179). The extracted DNA was dried for long-term storage at room temperature, using GenTegra®-DNA technology (Pleasanton, CA, USA). All samples used for molecular identification are stored in 2D SmartScan™ boxes at −80 °C (ABgene™, Portsmouth, NH, USA), each tube being equipped with a unique eight-digit code. These codes are linked to the occurenceID and eventID in the MEMO and MEMO+2020 datasets.

### MEMO and MEMO+2020 dataset creation

Collected data were entered into VecMap, exported and manually corrected by experts. A custom R & Grel (General Refine Expression Language) script was created to map the original data to Darwin Core as an event core with an occurrence extension [11]. Occurrence data from the MEMO (dataset 1) [2] and MEMO+2020 (dataset 2) [4] projects are extracted, standardised, and published as two separate Darwin Core Archives. The Darwin Core files are connected to the Belgian Biodiversity Platform IPT and documented with metadata. Datasets are published and registered with GBIF [3, 5].

The Darwin Core terms [22] in the dataset at the time of publication are as follows.

### Event core

```
id; eventID; type; language; license; rightsHolder; accessRights; datasetID;
institutionCode; datasetName; parentEventID; samplingProtocol; eventDate; habitat,
locationID; continent; countryCode; municipality; locality; decimalLatitude;
decimalLongitude; coordinateUncertaintyInMeters;geodeticDatum
```

### Occurrence extension

```
id; eventID; collectionCode; basisOfRecord; materialSampleID; occurrenceID;
recordedBy; individualCount; sex; lifeStage; establishmentMeans, occurrenceStatus;
identifiedBy; dateIdentified; identificationRemarks; scientificName; kingdom;
taxonRank; nomenclaturalCode
```

Object name: Darwin Core Archive MEMO+2020 - Monitoring Exotic MOsquitoes in Belgium

- DOI: https://doi.org/10.15468/r42fr7
- Character encoding: UTF-8
- Format version: 1.0
- Distribution: https://ipt.biodiversity.be/archive.do?r=itm-memoplus-occurrence
- Publication date of data: 2021-11-03
- Language: English
- Licences of use: https://creativecommons.org/publicdomain/zero/1.0/
- Metadata language: English



- Date of metadata creation: 2021-08-24
- Hierarchy level: Dataset

Object name: Darwin Core Archive MEMO-Monitoring Exotic MOsquitoes in Belgium

- DOI: https://doi.org/10.15468/4u5aub
- Character encoding: UTF-8
- Format name: Darwin Core Archive format
- Format version: 1.0
- Distribution: https://ipt.biodiversity.be/archive.do?r=itm-memo-occurrence
- Publication date of data: 2021-11-09
- Licences of use: https://creativecommons.org/publicdomain/zero/1.0/
- Language: English
- Date of metadata creation: 2021-08-24
- Hierarchy level: Dataset

## DATA AVAILABILITY

This data paper is linked with two MEMO-related datasets; dataset 1 [3] and dataset 2 [5]. The database server uses Windows Server 2003 SBS R2 as operating system, and is running IIS with PHP for site development, MS SQL Server for database development and SQL Server Mobile Tools to allow remote access from a PDA.

The generated microsatellites are deposited on the open-access Dryad Digital Repository [21]. The generated DNA barcodes are linked to NCBI BioProject ID: PRJNA837425.

The data are published under a Creative Commons CC0 waiver and we kindly ask you to notify the corresponding authors of the respective dataset if you use the data, especially for research purposes. Issues with the dataset can be reported at https://github.com/BelgianBiodiversityPlatform/data-publication-ITG/issues.

## AVAILABILITY OF SOURCE CODE AND REQUIREMENTS

- Project name: Repository for the publication of datasets from ITG
- Project home page: https://github.com/BelgianBiodiversityPlatform/data-publication-ITG
- Operating system(s): Platform independent
- Programming language: Jupyter, HTML
- Other requirements: none
- License: MIT.

## EDITOR'S NOTE

This paper is part of a series of Data Release articles working with GBIF and supported by the Special Programme for Research and Training in Tropical Diseases (TDR), hosted at the World Health Organization [23].

## DECLARATIONS
## LIST OF ABBREVIATIONS

EMS: exotic mosquito species; ITM: Institute of Tropical Medicine; MEMO: Monitoring of Exotic MOsquitoes; PoE: Point of Entry.



## ETHICAL APPROVAL

Not applicable.

## CONSENT FOR PUBLICATION

Not applicable.

## COMPETING INTERESTS

The authors declare that they have no competing interests.

## FUNDING

The MEMO and MEMO+2020 projects (2017–2020) were financed by the Flemish, Walloon and Brussels regional governments and the Federal Public Service (FPS) Public Health, Food Chain Safety and Environment in the context of the National Environment and Health Action Plan (NEHAP) (Belgium).

## AUTHORS' CONTRIBUTIONS

Principal investigators: WVB; ID
Resource contact, resource creator, point of contact: WVB; NS, ID
Metadata provider: DB
Content providers: WVB; ID, NS
Processors: DB.

## ACKNOWLEDGEMENTS

The authors would like to thank everybody who contributed to the creation of these datasets and this article, including the Belgian Biodiversity Platform [24], and GBIF [25]. This work is part of the MEMO and MEMO+2020 project funded by the Flemish, Walloon and Brussels regional governments and the Federal Public Service (FPS) Public Health, Food Chain Safety and Environment in the context of the National Environment and Health Action Plan (NEHAP) (Belgium). The Barcoding Facility for Organisms and Tissues of Policy Concern (BopCo [26]) is financed by the Belgian Science Policy Office (Belspo) as Belgian federal in-kind contribution to the European Research Infrastructure Consortium 'LifeWatch'. The Outbreak Research Team of the Institute of Tropical Medicine is financially supported by the Department of Economy, Science and Innovation of the Flemish government.

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
