## [Reviewer Report]

Upload additional filesDRR-202203-03/form/DRR-202203-03-Data-Review-CIH.pdfReviewer name and names of any other individual's who aided in reviewer Christopher HunterDo you understand and agree to our policy of having open and named reviews, and having your review included with the published papers. (If no, please inform the editor that you cannot review this manuscript.)YesIs the language of sufficient quality?YesPlease add additional comments on language quality to clarify if needed
Are all data available and do they match the descriptions in the paper? YesAdditional CommentsThere are a couple of minor discrepancies that should be addressed, please see my comments to the author.Are the data and metadata consistent with relevant minimum information or reporting standards? See GigaDB checklists for examples <a href="http://gigadb.org/site/guide" target="_blank">http://gigadb.org/site/guide</a>YesAdditional CommentsIs the data acquisition clear, complete and methodologically sound?YesAdditional CommentsThere is a minor issue with clarity of the number of sampling events at each location, see comments to the author for further details.Is there sufficient detail in the methods and data-processing steps to allow reproduction?YesAdditional CommentsIs there sufficient data validation and statistical analyses of data quality? YesAdditional CommentsIs the validation suitable for this type of data?YesAdditional CommentsIs there sufficient information for others to reuse this dataset or integrate it with other data?YesAdditional CommentsAny Additional Overall Comments to the AuthorThe manuscript describes “MEMO - Monitoring of Exotic MOsquitoes in Belgium is a sampling
event dataset”. Overall it is well written and the data provided to GBIF is well annotated and
appears to be mostly complete. There are no major problems that will prevent the publication of
this manuscript, but there are a number of minor points that require addressing first.
Number of GBIF datasets included in the manuscript - 2
Major comments (Author action required):
1 - Dataset 2 (MEMO+2020) in GBIF is currently listed as CC-BY license, this should be
updated to CC0, the text in the GBIF description for this dataset also needs updating with the
CC0 license.
2 - Both datasets appear to have multiple instances for the same location on the same day, my
assumption is that there were multiple traps, but this should be explicitly mentioned somewhere
in the manuscript.
See Curation checklist below for additional information.
Minor comments (Author action suggested):
1 - The text in the description of the dataset 1 (MEMO) in GBIF states “We have released this
dataset to the public domain under a Creative Commons Attribution 4.0 International (CC BY
4.0) license” - this needs to be changed to match the license specification in the metadata
(CCO).
2- This section from the manuscript mentions a great deal of additional data in passing, but
there are no links provided to those data points. “The caught EMS and five percent of all
collected mosquitoes were molecularly identified to validate and confirm the morphological
identification. Tissue and DNA were subsequently deposited in a molecular reference collection
hosted at the Royal Belgian Institute of Natural Sciences (RBINS). A specific molecular
identification pipeline was developed to enable the accurate identification to species (or biotype
for Culex pipiens) level of all mosquitoes occurring in Belgium (native and potential EMS).
Further, a morphological collection with a representation of 23 species and the most intact
specimens sampled during the MEMO project was generated for future reference and is also
hosted at RBINS.”
2a. While the citation of the manuscripts that contain the NCBI sequence data is technically
sufficient, I would prefer the NCBI BioProject accession number of the sequence be provided
here.
2b. The accessions of the physical specimens hosted at RBINS should be included in a
supplemental table. - I have now seen this is already provided later in the MS, thank you.
2c. The pipeline code used to process the molecular data should be made available,
preferably via GitHub.
3 - There is a slight discrepancy between the dates mentioned in the manuscript and dataset
metadata verses the actual event dates in the dataset data for both datasets. Please check the
dates carefully and adjust accordingly in the manuscript and dataset metadata.
4 - There is a discrepancy between the number of sampling locations mentioned in the
manuscript (20-23) and the actual number of different locations where sampling events occurred
(33), please clarify the sampling locations used in the manuscript and include all sampling
locations in the Figure 1 map.
5- Figure 2 needs to be adjusted to match the data available in GBIF.
See Curation checklist below for additional information.RecommendationMinor Revision

---

## [Reviewer Report]

Reviewer name and names of any other individual's who aided in reviewer Constantino González-SalazarDo you understand and agree to our policy of having open and named reviews, and having your review included with the published papers. (If no, please inform the editor that you cannot review this manuscript.)YesIs the language of sufficient quality?YesPlease add additional comments on language quality to clarify if needed
Are all data available and do they match the descriptions in the paper? YesAdditional CommentsAre the data and metadata consistent with relevant minimum information or reporting standards? See GigaDB checklists for examples <a href="http://gigadb.org/site/guide" target="_blank">http://gigadb.org/site/guide</a>YesAdditional CommentsIs the data acquisition clear, complete and methodologically sound?YesAdditional CommentsIs there sufficient detail in the methods and data-processing steps to allow reproduction?YesAdditional CommentsIs there sufficient data validation and statistical analyses of data quality? NoAdditional CommentsThe authors do not perform any type of statistical analysis. An analysis of the sampling effort performed is suggested to the authors to determine the completeness of the sampling. This type of analysis will allow identifying those sites that will require a sampling effort in future research.Is the validation suitable for this type of data?NoAdditional CommentsIs there sufficient information for others to reuse this dataset or integrate it with other data?YesAdditional CommentsAny Additional Overall Comments to the AuthorRecommendationAccept